# Maturation of Aluminium Adsorbed Antigens Contributes to the Creation of Homogeneous Vaccine Formulations

**DOI:** 10.3390/vaccines11010155

**Published:** 2023-01-11

**Authors:** Donatello Laera, Camilla Scarpellini, Simona Tavarini, Barbara Baudner, Agnese Marcelli, Carlo Pergola, Malte Meppen, Derek T. O’Hagan

**Affiliations:** 1Technical & Research Development, Drug Product, GSK, 53100 Siena, Italy; 2Laboratory of Medicinal Chemistry, University of Antwerp, Universiteitsplein 1, 2610 Antwerp, Belgium; 3Bacterial Vaccine R&D, GSK, Cellular Immunology and In-Vivo Models, IT Immunology, 53100 Siena, Italy; 4Technical & Research Development, Drug Product, GSK, Rockville, MD 20850, USA

**Keywords:** aluminium hydroxide, adsorption, distribution, maturation, ageing, populations, homogeneity, compounding strategy

## Abstract

Although aluminium-based vaccines have been used for almost over a century, their mechanism of action remains unclear. It is established that antigen adsorption to the adjuvant facilitates delivery of the antigen to immune cells at the injection site. To further increase our understanding of aluminium-based vaccines, it is important to gain additional insights on the interactions between the aluminium and antigens, including antigen distribution over the adjuvant particles. Immuno-assays can further help in this regard. In this paper, we evaluated how established formulation strategies (i.e., sequential, competitive, and separate antigen addition) applied to four different antigens and aluminium oxyhydroxide, lead to formulation changes over time. Results showed that all formulation samples were stable, and that no significant changes were observed in terms of physical-chemical properties. Antigen distribution across the bulk aluminium population, however, did show a maturation effect, with some initial dependence on the formulation approach and the antigen adsorption strength. Sequential and competitive approaches displayed similar results in terms of the homogeneity of antigen distribution across aluminium particles, while separately adsorbed antigens were initially more highly poly-dispersed. Nevertheless, the formulation sample prepared via separate adsorption also reached homogeneity according to each antigen adsorption strength. This study indicated that antigen distribution across aluminium particles is a dynamic feature that evolves over time, which is initially influenced by the formulation approach and the specific adsorption strength, but ultimately leads to homogeneous formulations.

## 1. Introduction

Active immunization using prophylactic vaccines is one of the most efficient ways of protecting the global population against infectious diseases and life-threatening epidemics [1]. The use of recombinant protein subunits has long been preferred over attenuated pathogens for reasons of safety and ease of production [2]. Antigens are often not sufficiently immunogenic, leading to the use of adjuvants in vaccine formulations to boost immunogenicity [3,4,5]. The most widely used adjuvant system is a suspension of aluminium oxyhydroxide (here referred to as AlumOH), a salt which exists as an aggregate of particles ranging from 1 to ~20 µm, with a Point of Zero Charge (PZC) of 11.4 and thus a positive charge at neutral pH [6].

Although human vaccines adjuvanted with AlumOH have been used successfully since 1926, their mechanism of action is still not fully understood [7]. Some studies have proposed that antigen adsorption to the inorganic particles prior to injection is a key factor towards the ‘adjuvant effect’ of aluminium salts that promote antigen uptake by antigen presenting cells (APC) [8]. Meanwhile, other studies have suggested that the mere presence of AlumOH induces immunopotentiation, and that the adsorption of the antigen to AlumOH and retention in the injection site is not required [9,10,11]. Nevertheless, since 1977 the World Health Organization (WHO) has recommended that in vaccines containing aluminium-adjuvants, antigen adsorption should be maximized ≥80% [12]. For AlumOH-based vaccines, the degree of antigen adsorption therefore represents an important parameter to control, and is an indicator of the consistency of respective vaccine formulations and the production processes [13]. In terms of interactions at the molecular level between the aluminium adjuvant and the respective protein antigen(s), however, dedicated studies remain scarce. Thus, to increase our understanding of AlumOH-adjuvanted vaccines, it is of the utmost importance to accumulate data about characterization topics such as: the distribution of antigens on the surface of the bulk AlumOH; the actual antigen structure upon adsorption; and epitope orientation. Currently, only a limited number of cutting-edge techniques have been developed in this regard [14,15,16,17]. To better understand the mechanism of action of adjuvanted vaccines, therefore, it is essential to directly characterize the complex construct resulting post adsorption of (several) antigens onto the AlumOH surface [18]. Flow cytometry (FC) [14,19,20] has previously been used to understand the behaviour of antigens when formulated in adjuvanted vaccines without an antigen desorption procedure required prior to sample analysis. This direct characterization at the molecular level becomes highly challenging when dealing with combination vaccines, where several antigens are co-adsorbed onto AlumOH, and the developed techniques need to be antigen specific and robust. There are several factors affecting AlumOH-antigen interactions, and among these are: pH [21]; ionic strength [22]; PZC of AlumOH; and the isoelectric point (IEP) of the antigen [23,24], which is particularly important for electrostatic interactions between aluminium salts and antigens with an opposing charge. Additional parameters pertain to the order of addition of excipients [25] (important for hydrophobic and ligand exchange interactions), the molecular weight of the antigen, and the size of the AlumOH particles [26]. The role of the formulation process on the antigen interaction with the AlumOH particle is therefore an important parameter to take into account when preparing a multivalent formulation [27]. Currently, three main strategies (Figure 1) are used to produce combination vaccines: sequential adsorption (antigens A, B, C and D are added one after the other onto AlumOH); competitive adsorption (antigens A, B, C and D are formulated simultaneously, i.e., antigen mixture prior to AlumOH adsorption); and separate adsorption (antigens A, B, C and D are adsorbed individually onto AlumOH and then pooled).

There are multiple points to consider when executing different formulation procedures that relate to: the binding of the antigens to AlumOH; the dose uniformity of each single component; antigen distribution across the AlumOH bulk; and the stability of the mixed product. Little data has so far been published on how antigens are distributed at the surface of AlumOH particles, and their trend to reach uniformity after respective processes [15]. The goal of this study was to establish and monitor the interaction of four adsorbed antigens using a model AlumOH-adjuvanted tetravalent vaccine (AlOH-4Ag). Three different formulation approaches were used, and the formulation samples were monitored regularly for 130 days. The antigens employed in this study were four recombinant proteins selected for their chemical-physical properties (Table 1); different isoelectric points, representative of varied electrostatic interaction strengths [28]; and comparable molecular weights [26].

The three major formulation strategies mentioned above (Figure 1) were used for the preparation of identical AlOH-4Ag vaccines in terms of adjuvant concentration and antigens dose. The following main objectives were investigated:

Can the different formulation approaches (Figure 1), together with different electrostatic strengths (Table 1), affect the antigens distribution over surface of AlumOH particles?

If this is the case, does the storage time influence antigen distribution, and how?

What is the minimum time required to reach uniform distribution of antigen-AlumOH complexes?

Samples derived from the three formulation approaches (sequential in blue, competitive in green and separate in red) were monitored for attributes relevant for the scope of the study, at different times. Following this approach, we were able to distinguish between the adsorption mechanism caused by strong, intermediate, or weak electrostatic interactions depending on the isoelectric point of each antigen. Importantly, we were able to monitor how the interaction between each antigen with the AlumOH adjuvant varies over time depending on the formulation strategy used. We showed that antigen redistribution on the surface of AlumOH results in homogeneous adjuvant–antigen complexes, independently of the formulation strategy used.

## 2. Materials and Methods

### 2.1. Antigens

Research-grade material antigens used in this study were provided by GSK Pre-Clinical/Antigen Design Italy: all antigens were in 10 mM KH_2_PO_4_, pH 7.

### 2.2. Adjuvants, Buffers and Excipients

GSK manufactured AlumOH was used in this study. Buffers and excipients were prepared by TRD-Drug Product, Italy using: L-Histidine (Sigma, St. Louis, MO, USA, HS034), NaCl (Sigma, S7653) and HyPure WFI Quality water (Hyclone, Logan, UT, USA, SH30221).

### 2.3. Preparation of Formulation Samples

#### 2.3.1. Sequential Adsorption Approach

AlumOH was added to a reaction vessel in the presence of buffer and excipients, and antigens were then added sequentially, according to the following order: A, B, C and D. The formulation approach was considered complete after the addition of last antigen in the sequence. 5 mL of sample was prepared at day 0: the AlumOH-based sample was prepared by mixing 200 µg of AlumOH with 10 µg of each antigen in 10 mM histidine buffer (pH 6.5), final formulation volume of 100 µL. The osmolarity was adjusted to 300 mOsm/kg using a 2 M NaCl solution. The sample was stirred at room temperature (RT) for 1 h.

#### 2.3.2. Competitive Adsorption Approach

Three mL of two different samples were prepared at day 0: in the first sample all antigens were present at a concentration of 20 µg in 100 µL, (a 2-fold concentration with respect to the final dosage), in a 10 mM histidine buffer (pH 6.5) which was then adjusted to 300 mOsm/kg using a 2 M NaCl solution. The second sample contained an AlumOH suspension at a concentration of 400 µg in 100 µL, (again a 2-fold concentration with respect to the final dosage), in 10 mM histidine buffer (pH 6.5) and adjusted to 300 mOsm/kg using a 2 M NaCl solution. An equal volume of each sample was then mixed for a minimum of 1 h at RT.

#### 2.3.3. Separate Adsorption Approach

Each of the four antigens was formulated separately in the presence of AlumOH in monovalent independent vials at day 0:2 mL of each formulation sample was prepared by mixing 200 µg of AlumOH with a single antigen at a concentration of 40 µg in 100 µL, (a 4-fold greater concentration with respect to the final dosage), in 10 mM histidine buffer (pH 6.5) and adjusted to 300 mOsm/kg using a 2 M NaCl solution. These (monovalent) formulation samples were incubated for a minimum of 1 h at RT, followed by the mixing of equal volumes of each monovalent formulation to obtain the tetravalent formulation sample for a minimum of 1 h at RT.

### 2.4. pH-Metry

pH was measured using a Cyberscan pH 1500 pH-meter (Eutech Instruments Europe BV). A volume of 50 µL/formulation was used for the analysis.

### 2.5. Sodium Dodecyl-Sulfate PolyAcrylamide Gel Electrophoresis (SDS-PAGE)

The AlumOH adsorption capacity was evaluated for antigen A, B, C and D in the multivalent vaccine formulations prepared with different compounding strategy. To determine antigen adsorption, each formulation was centrifuged at 3000× *g* for 10 min at room temperature (RT) and the supernatant was removed without disturbing the pellet. To precipitate and concentrate the possible unbound antigen, the supernatants were treated with 0.5% deoxycholate sodium salt and incubated for 10 min at RT, followed by the addition of 60% trichloroacetic acid (TCA). The TCA pellets were resuspended with loading sample buffer (LSB) from ThermoFisher Scientific (Waltham, MA, USA), P/N NP0007, whereas the aluminium pellets were reconstituted with desorption buffer containing LSB with 0.5 M sodium phosphate at pH 6.8. The TCA-treated samples, aluminium pellets, and standard controls (antigens at 1, 0.5, and 0.25 µg) were heated at 95 °C for 10 min and loaded into a NUPAGE 4–12% gradient Bis–Tris Midi gel from ThermoFisher Scientific, P/N WG1401BOX, and run under reducing conditions in MOPS 1X SDS running buffer from ThermoFisher Scientific, P/N B0001, at a constant voltage of 200 V for approximately 50 min [14]. The gel was stained using SimplyBlue™ SafeStain from ThermoFisher Scientific, P/N LC6065, according to provider’s instructions.

### 2.6. Static Light Scattering (SLS)

The particle size distribution (PSD) of each AlumOH sample was measured with the Beckman Coulter Laser Diffraction Particle Size Analyzer LS13320, according to the manufacturer’s instructions. A volume of around 500 µL/formulation was placed in the cell of a micro-liquid module (MLM) with ~12 mL of formulation buffer (10 mM histidine buffer pH 6.5, 150 mM NaCl). For each sample, three independent measurements of 90 s at room temperature were acquired. As a negative control, a sample containing plain AlumOH at 2 mg/mL was used.

### 2.7. Zeta Potential (ZP)

The surface charge of AlumOH particles was measured through zeta potential analysis using a Malvern Zetasizer Nano ZS90, according to the manufacturer’s instructions. A volume of 750 µL was placed in a disposable folded capillary cell (DTS1070) purchased from Malvern. Each formulation sample was read in triplicate, with three scans per each reading, after 90 s equilibration time at 25 °C. From each formulation, with a final concentration of 2 mg/mL of AlumOH and 10 µg/100 mL of antigens, an aliquot of 93.75 µL was diluted with 656.25 µL of formulation buffer (10 mM Histidine buffer pH 6.5, 150 mM NaCl). As the negative control a sample containing plain AlumOH at 2 mg/mL was employed.

### 2.8. Flow Cytometry (FC)

Briefly, the FC technique has been accordingly adapted to characterize particulate vaccine containing Alum-based adjuvants: in particular, it allows for the detection, at the same time, both single AlumOH particles (morphological analysis) and antigen(s) delivered on their surface (fluorescence analysis, through a double-staining step using an antigen-specific primary antibody and a secondary labelled antibody) [14]. Specific polyclonal rabbit antibodies against each single antigen were generated by immunization of rabbits with purified recombinant proteins.

### 2.9. Separate Adsorption Sample Sorting and Western Blot Analysis

A major application of FC is to separate events according to specific population features for further and more accurate studies: this process is called sorting or fluorescence-activated cell sorting (FACSTM) analysis. In this study, we applied FACSTM analysis to specifically sort AlumOH particles with different amounts of adsorbed antigen A (based on flow cytometry analysis, see results Section 3.4.3.) in order to confirm, through an orthogonal technique, if these differences effectively reflect different antigen content delivered on AlumOH populations. AlumOH particles of the separate adsorption samples at T0 were stained with polyclonal rabbit antibodies specific for antigen A, followed by a goat anti-rabbit secondary antibody, AlexaFluor 647 IgG (H + L) fluorescently labelled, and incubated. The dilution and buffer used were the same as those applied for the FC analysis. SO-stained particles were then examined using BD FACSAria III Cell Sorting System with BD FACSDiva Software (BDBiosciences). After morphology and singlets selection, the remaining gated particles were sorted based on two distinct population behaviors: respectively, the first with the lower and the second with the higher mean fluorescence intensity (MFI) for antigen A. The AlumOH particles were sorted through a 100 μm nozzle at a sheath pressure of 20 psi and a highly pure sorting modality (2-way 0-32-0 sorting) was chosen. The flow rate during the sorting was approximately 800/1000 events/second. The two SO-sorted populations (each about 1*10^6^ particles) were collected in 5 mL polypropylene tubes containing 500 µL of formulation buffer. This sorting procedure was repeated four times, in order to have an equal volume for further and independent Western blot (WB) analysis of all antigens formulated (A, B, C and D). Immediately after collection, the particles were concentrated using 0.5% deoxycholate sodium salt (DOC) and 60% trichloroacetic acid (TCA), followed by centrifugation at 20,000× *g* for 10 min. Subsequently, the supernatants (SNs) were poured off and the AlumOH precipitates were re-suspended in 100 µL of Sodium Dodecyl-Sulfate (SDS) desorption buffer (900 mM Na_2_HPO_4_, pH 6.8, in presence of DTT) and heated at 95 °C for 10 min. Each of two sorted populations (respectively, for antigen A, C and B) were, together with appropriate controls (standard antigen amounts at 100, 50, 25 and 12.5 ng), loaded into a NUPAGE Novex 4–12% gradient Bis–Tris Mini gel (Invitrogen, Waltham, MA, USA) and run under reducing conditions at a constant voltage of 200 V for approximately 50 min. Since the sorting procedure was repeated four times, the so sorted populations were loaded in four different SDS-PAGEs. Subsequently, all gels were transferred to nitrocellulose membranes using an iBlot Gel Transfer Device (ThermoFisher Scientific). After transferring, all membranes were saturated with blocking buffer (5% milk powder and 0.1% Tween20 in 1X PBS) for 1 h at RT, followed by overnight incubation at 4 °C with primary polyclonal rabbit against antigen A, B, C and D diluted 1:1000 in blocking buffer, each one, respectively, used for one membrane. After three washes with blocking buffer, the four membranes were incubated for 2 h at RT with an anti-rabbit immunoglobulin G (IgG), horseradish peroxidase (HRP)-linked antibody from a tab (GE Healthcare, Chicago, IL, USA, AP510P), diluted 1:10,000 in blocking buffer. Following further washes, the blots were developed by incubation with Enhanced ChemiLuminescence (ECL™) blotting reagents (GE Healthcare, RPN2109) according to the manufacturer’s instructions. Films were exposed at different times starting from 1 to 5 min.

## 3. Results

The methods indicated in Table 2 were selected to provide information on the characteristic attributes deemed important for this investigation: buffer (i.e., pH); adjuvant (i.e., AlumOH size and charge); antigen (i.e., integrity); and adjuvant–antigen interaction (i.e., antigen adsorption and distribution). The pH of all formulations used in this work was maintained constant by using a histidine buffer (6.5 ± 0.5) in an isotonic medium (150 mM NaCl).

### 3.1. pH Values and Particle Size Distribution of Formulation Samples

pH values (Figure 2) were within the range of 6.5–6.7 during the entire kinetic study for all three formulation approaches.

Figure 2, 8 µm (number % distribution analysis) and a standard deviation of about 1.4 µm (Figure 3). An antigen-free sample of AlumOH at 2 mg/mL, in 10 mM histidine buffer (pH 6.5) and 150 mM NaCl was used as control for PSD analysis and displayed a size around 2.4 µm (data not shown). Formulation samples after 50 and 130 days were not assayed for PSD, in order to limit sample consumption since the assay required the largest volume (about 500 µL).

### 3.2. Antigen Integrity and Adsorption onto AlumOH

Each formulation approach displayed similar antigen integrities and adsorption profiles (Figure 4), both at time point 0 and after 100 days. For each formulation, all antigens conformed to standard controls and no molecular weight degradation profiles were detected.

AlumOH adsorption profiles for all antigens displayed similar trends. Antigen C was the only antigen not completely adsorbed, as traces were detected in the supernatant post-centrifugation and separation from the AlumOH pellet and tri-chloro acetic (TCA) precipitation steps.

Results obtained for both antigen integrities and adsorption profiles at remaining time points (see Appendix A) agreed with those achieved at time point 0 and 100 days. Formulation samples after 130 days were not tested.

### 3.3. Surface Charge of Different Formulation Samples

Similar trends were observed for ZP results throughout the kinetic study for each formulation approach used (Figure 5). All formulation samples displayed values that moved from around 0 at the starting point, falling to −10 mV after 30 days. From 50 to 130 days, all formulation samples showed little change in their zeta potential, with values within the range of −3/−5 mV. These dynamic trends reflect changes that are occurring across the bulk AlumOH surface due to an overall shift caused by adsorbed antigens, with no possibility of discriminating the contribution of each single component. A sample of AlumOH at 2 mg/mL, in 10 mM histidine buffer (pH 6.5) and 150 mM NaCl, in absence of antigens, was used as control and displayed a positive value around 22 mV (data not shown).

### 3.4. Antigens Distribution over AlumOH Particles through FC Analysis

In Figure 6, Figure 7 and Figure 8 histograms referring to the distribution of each antigen on AlumOH are displayed after flow cytometry analysis (the strength of the electrostatic interaction decreases from left to right).

#### 3.4.1. Time Point 0 and 100/130 Days

At (T0), each formulation approach used led to a different antigen distribution, with sequential and competitive approaches showing similar profiles that clearly differ from the separate formulation strategy (Figure 6). In the case of sequential (blue) and competitive (green) adsorption samples, model antigen B, C and D displayed a similar, narrow, and mono-dispersed AlumOH-antigen complex population, whereas antigen A displayed a broader and heterogeneous population in sequential adsorption. In the vaccine sample prepared through separate (red) adsorption approach, all antigens exhibited a highly poly-dispersed population. Antigen A, B and C displayed two separate peaks (with discernible fluorescence intensities), while antigen D had a unique, wide population with two minor additional sub-populations, one on the left and one on the right. Importantly, over time a redistribution was observed in all formulations independently of the adsorption approach used: an overlay of the datasets obtained from the sequential and competitive adsorption approaches after 100 days with that from the separate adsorption approach after 130 days (Figure 6) resulted in an almost equal distribution for all antigens over AlumOH particles, regardless of electrostatic strength. These findings demonstrate that separate adsorption formulation procedures reach a uniform antigen distribution after an extended period of time.

In order to better understand the behaviour of each antigen when adsorbed to AlumOH separately, we next compared the fluorescence spectra obtained using the separate formulation approach with a negative control, as well as monovalent formulation (Figure 7). It was found that:

For the model antigens A, B and C there was an overlay of each monovalent sample with the higher fluorescence intensity peak of the separate adsorption sample; for the model antigen D, an overlay with the monovalent sample occurred with the minor additional population on the right of the unique, wide population.

As expected, the negative control sample overlapped (partially or totally) with the lower fluorescence intensity peak of separate adsorption sample for antigens A, B and C; for antigen D, overlay of the negative control occurred with the minor additional population on the left of the unique, wide population.

#### 3.4.2. Overlay of Different Time Points for Separate Adsorption Approach Sample

To more clearly see the evolution of antigen adsorption over time, the data obtained for each antigen (separate adsorption) at T0, T30 (or T50) and T130 were compared (Figure 8). Antigen A displayed two separate peaks at time point 0 (dark red), one on the left and one on the right, with low and high fluorescence intensity, respectively. After 30 days of incubation (green), the peak on the left shifted to the right (with an increased fluorescence intensity), while the second peak maintained a similar fluorescence intensity value with respect time point 0. After 130 days (light violet), the AlumOH-antigen A complex showed a unique and mono-dispersed population that overlapped with the higher fluorescence intensity peak of time 0 and 30 days. Similar to antigen A, antigens B and C displayed two separate peaks at time point 0; after 30 days of incubation for antigen B (50 days for antigen C), the peak on the left shifted on the right and merged with the second peak, resulting in a unique, yet poly-dispersed, AlumOH-antigen population. After 130 days, the AlumOH-antigen complex for antigen B and C showed a unique and mono-dispersed population that overlaps with the higher fluorescence intensity peak observed at time 0. As opposed to the other antigens tested, antigen D displayed a unique, poly-dispersed population at time point 0 with two minor populations, one on the left and one on the right with respect to the main peak, with lower and higher fluorescence intensities, respectively. After 30 days of incubation, these three populations merged into a unique peak resulting in a mono-dispersed AlumOH-antigen C population which overlaps with the minor population (with higher fluorescence intensity) detected at time point 0 on the right. After 130 days, the AlumOH-antigen C complex showed an even more mono-dispersed population with respect to the 30 day incubation.

#### 3.4.3. Sorting of Separate Adsorption Sample and Western Blot Analysis for Antigen A

Two AlumOH-antigen A populations from separate adsorption samples (red) at T0 were selected for sorting through FACS analysis by overlaying (Figure 7) with a negative control (light blue) and antigen A monovalent peaks (green). They were expected to deliver almost, respectively, none (named P1) and all (named P2) amount of the corresponding antigen A. Approximately 1*10^6^ AlumOH particles for both sorted populations were collected; later samples were treated for further SDS-PAGE/Western blot (WB) analysis. Since the sorting procedure was repeated in the same way four times, samples were loaded in four different gel electrophoresis for immuno-blotting, and incubated respectively with antigen A, B, C and D polyclonal rabbit sera. It was therefore possible to evaluate not only the content of antigen A delivered from each sorted population, but also those of other antigens. The results of the WB analysis for antigen A are displayed in Figure 9.

WB analysis after incubation with antigen A sera (Figure 9) of the sorted AlumOH populations P1 and P2 clearly confirmed the flow cytometry results (Figure 7): indeed, antigen A was detected only in the P2 population (Figure 9, WB, well 8). This result indicates that the antigen A at T0, due to its strong electrostatic interaction, remains mostly adsorbed on the AlumOH population P2, which derives from the antigen A monovalent sample of separate adsorption sample (see overlay in Figure 7). On the other hand, at T0 no traces of antigen A were detected in P1 population (Figure 9, WB, lane 7). This evidence means that no redistribution of the antigen A from P2 population on remaining AlumOH population P1 occurs immediately. It is noteworthy to mention that AlumOH population P1 derives from the monovalent sample B, C and D used to prepare multivalent separate adsorption sample. It is instead interesting to underline how antigen B, C and D were detected in both populations with a similar content (see Appendix A). This behaviour can be explained since both P1 and P2 populations of separate adsorption sample at T0 were sorted after specific antigen A staining at flow cytometry, and, indeed, they reflected the specific distribution only for antigen A.

## 4. Discussion

From a drug product development perspective, the most important activities for an optimal vaccine design can be summarized in three main pillars: screening of the most appropriate formulation buffer excipients to ensure stability of the antigen(s) and the adjuvant at an early phase project [29,30,31]; selection of the relevant attributes of antigen(s) to be monitored through execution of different types of assays [12,32,33]; and identification of more advantageous and suitable compounding procedures during process development steps. While the importance of the first two pillars is now commonly accepted and recognized among the scientific community, the compounding strategy remains poorly developed, in particular for AlumOH-based vaccines where antigen adsorption onto adjuvant particles has a key role [8]. The reason for this can partly be explained by the use of conventional vaccine characterization assays, which mostly focus both on physico-chemical parameters such as pH [21], antigen identity and stability [34], analysis of specific adjuvant property [23,26] and on biological testing as for sterility and immunogenicity (in vivo) or antigenicity (In Vitro) [35]. Additionally, the use of in vitro science-based assays (reliable, faster and aligned with the principles of replacement, reduction and refinement (3R) of animal testing) has grown rapidly in recent years [36]. In vitro assays are often antibody-based and can recognize specific and/or conformational epitopes to be used as a stability indicator [17]. These assays are also able in the presence of AlumOH to detect the mode and site of antigen binding [37]. Currently, however, in the literature there is a scarcity of articles dedicated towards furthering the understanding of the aspects that impact AlumOH-antigen complexes, such as antigen distribution, orientation, and structure in multivalent formulation. For this purpose, we investigated three different approaches (Figure 1) for formulating an AlumOH-adjuvanted tetravalent vaccine model (AlOH-4Ag) in order to explore if different formulation approaches, together with the antigen-specific electrostatic adsorption strengths, can affect the antigen distribution across AlumOH particles and to what extent the storage/incubation times influence this phenomenon. This approach also allowed us to investigate if there is a minimum time range to reach uniform antigen distribution. Four representative antigens were selected for this investigation with comparable molecular weights [26], ranging from about 23 to 34 kDa, but also having diverse IEP values which lead to supposedly different adsorption strengths (Table 1, i.e., from strong (antigen A) to weak (antigen D)) [28]. The formulation approaches differed only in terms of the method of addition between the antigens and the AlumOH adjuvant, namely, antigens were either sequentially added onto AlumOH (sequential adsorption, highlighted in blue), pooled and then adsorbed all together to the adjuvant (competitive adsorption, highlighted in green) or individually adsorbed to AlumOH followed by mixing of the monovalent formulations (separate adsorption, highlighted in red) [27]. Importantly, all three formulation samples were normalized in terms of formulation buffer composition (10 mM Histidine pH 6.5, 150 mM NaCl), and after formulation they were stored at 2/8 °C without any further mixing that could lead to changes of the antigens at the AlumOH surface. All formulations resulted in: comparable values of pH (Figure 2) and AlumOH PSD (Figure 3); similar antigen integrities and adsorption rates (Figure 4); and comparable trends during storage time for surface charge values (Figure 5). Despite the different compounding strategy used for each sample, no major differences were observed during the storage of these formulation samples using physico-chemical parameters and conventional analytical approaches: as a consequence, we concluded that stability should not be an issue. Indeed, changes in pH that could induce both antigen desorption from AlumOH [21] and antigen chemical instability were not detected [38]. At the same time any AlumOH particle aggregation that could cause protein desorption was not measured, as the total particle surface area would decrease with increasing particle size [26]. It was however useful to apply FC, based on immuno-detection [14], to provide insight into the properties of each AlumOH particle, focusing simultaneously on both the adjuvant and the adsorbed antigen. In our case it was evident that, at T0, the separate adsorption sample (Figure 6, red histogram) displayed high poly-dispersity in terms of antigen distribution across AlumOH particles. The use of an orthogonal approach (Western blot analysis, after a sorting step through FACS technique) confirmed this observation (Figure 9), meaning that AlumOH particles effectively delivered different content of adsorbed antigen A. The dispersity could be explained in the first instance by the formulation strategy used to prepare samples: indeed, from the very beginning, the separate adsorption samples have always displayed, for all antigens, different AlumOH populations with respect to sequential (blue) and competitive (green) samples, which instead displayed mono-dispersed AlumOH-antigen complexes. Secondly, electrostatic interaction strengths also had a role in determining the level of poly-dispersity in the separate adsorption samples, with antigen A (strong electrostatic interaction) and antigen D (weak electrostatic interaction) displaying contrasting behaviour: two well-defined peaks for antigen A, and a unique main population between two smaller subpopulations for antigen D. An overlay (Figure 8) of different time points for separate formulation approach clearly showed that, proportional to specific antigen electrostatic strengths, each antigen was changing distributions over time, approaching dose uniformity at each single particle: antigen D and antigen A, respectively, with a faster (~30 days) and slower (~130 days) behaviour. Finally, independent of the compounding strategy used, all formulation samples reached an equal antigen distribution across AlumOH particles, with a longer incubation time required for the separate approach with respect to sequential and competitive procedures (Figure 6). Considering that no mixing procedures were carried out during storage at 2/8 °C in this study, a possible explanation to these phenomena could be that antigens experience an equilibrium based on continuous desorption and re-adsorption at the AlumOH surface through multiple noncovalent and reversible interactions (H-bonding, Van der Waals Forces, electrostatic and hydrophobic interactions) favoured by the many functional groups that are present within the antigens [26,39,40]. Considering the controversial role of antigen adsorption and depot effect [9,10,11] in inducing AlumOH immunopotentiation, and that electrostatic interactions are weaker with respect to ligand exchange adsorption [8] leading to possible elution of antigen from AlumOH in the presence of interstitial fluid [41], we suggest that any antigen distribution over AlumOH particles should not be considered a critical quality attribute that affects vaccine performance in vivo. Moreover, from a vaccine manufacturing point of view, it is still important to maintain a high antigen adsorption rate for consistency of drug product processes [13] and to protect antigens from thermal degradation, since it preserves antigen stability [42,43]. The choice of a compounding procedure with respect to another should therefore be driven by pragmatic advantages highlighted during the process development/scale up activities of late development phase and/or commercial launch campaign. For example, the use of individually pre-adsorbed antigen bulks to be later used in different multivalent drug product vaccine formulations can represent a manufacturing benefit. In addition, pooling together all the antigens prior to AlumOH addition would be useful from a sterility assurance point of view since it would reduce the number of 0.22 µm filtration steps (needed for each antigen) required. All formulation strategies here described for preparing an AlumOH-based vaccines can be used interchangeably during the preclinical and/or early development phases since different antigen distribution should be not a parameter to take into account when performing in vivo animal experiments for the reasons discussed above. On the contrary, when immuno-based in vitro assays are used to characterize product performance and increase product understanding more in-depth, the aging effect and formulation strategy used to prepare AlumOH-based vaccines should be taken into account to properly interpret results. For this reason, the use of a sequential or competitive approach is recommended during the preclinical and/or early development phases because they represent the most straightforward approach to guarantee a high level of homogeneity in terms of antigen distribution across AlumOH particles.

## 5. Conclusions

In this study, we tested three different adsorption approaches (sequential, competitive, and separate) to prepare an AlumOH-based tetravalent protein vaccine. Formulation samples differed only in terms of the compounding procedure followed, while excipient concentrations and doses were identical. The antigens selected covered a range of strong and weak electrostatic strengths. Kinetic studies demonstrated that all AlumOH-based vaccines behaved in the same way: pH values were stable, AlumOH PSD and antigen adsorption levels were similar for all formulations. The surface charge values calculated across the whole formulation highlighted similar trends, whilst also suggesting that changes were occurring over time. These results were confirmed through FC, which demonstrated that the formulation strategy had an active role in causing different antigen distribution across AlumOH particles, and that the time needed to reach a uniform AlumOH–antigen complex is a function of the specific antigen electrostatic strength.

## Figures and Tables

**Figure 1 vaccines-11-00155-f001:**
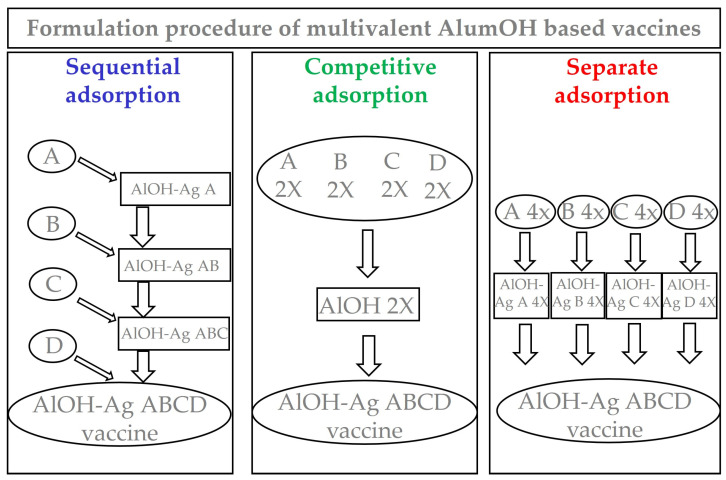
Figure describing three different formulation approaches for aluminium-based vaccines. The values expressed as 2X in competitive adsorption refers to antigens and AlumOH content of intermediate drug product, while those 4X in separate adsorption refers only to antigens content of monovalent bulks. In all formulation samples, the final drug products have exactly same content of both AlumOH and antigens.

**Figure 2 vaccines-11-00155-f002:**
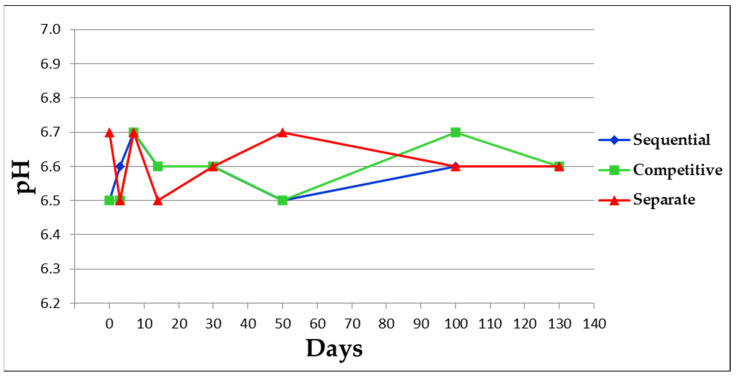
pH value of each single formulation sample at different time points.

**Figure 3 vaccines-11-00155-f003:**
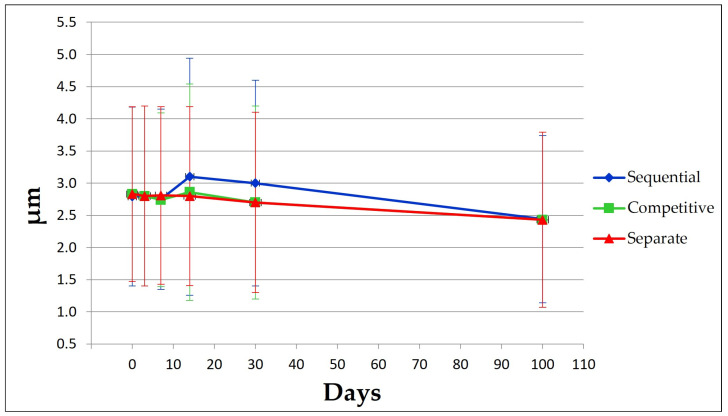
PSD results at different time points for each sample. PSD results are expressed as the mean value of three independent measurements with standard deviation.

**Figure 4 vaccines-11-00155-f004:**
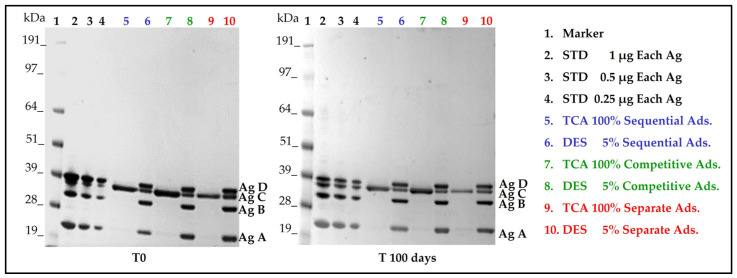
SDS-PAGE results at time points 0 and 100 days which show antigens identity in respect to standard controls and degree of antigens adsorption after steps of centrifugation and separation of supernatants from AlumOH pellet. STD: Standard antigens solution of known amount; Ag: Antigen; TCA: Supernatant precipitated with tri-chloro acetic acid after steps of centrifugation and AlumOH separation; Ads: Adsorption; DES: AlumOH desorbed with desorption buffer following steps of centrifugation and supernatant separation.

**Figure 5 vaccines-11-00155-f005:**
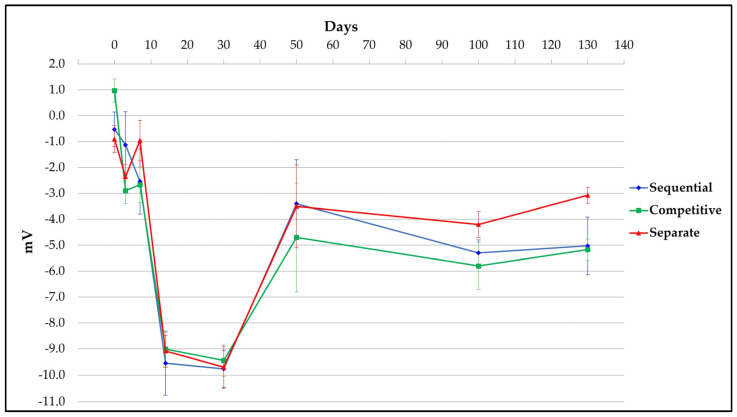
Mean and standard deviation of ZP value for three different replicates of each single formulation sample at different time points.

**Figure 6 vaccines-11-00155-f006:**
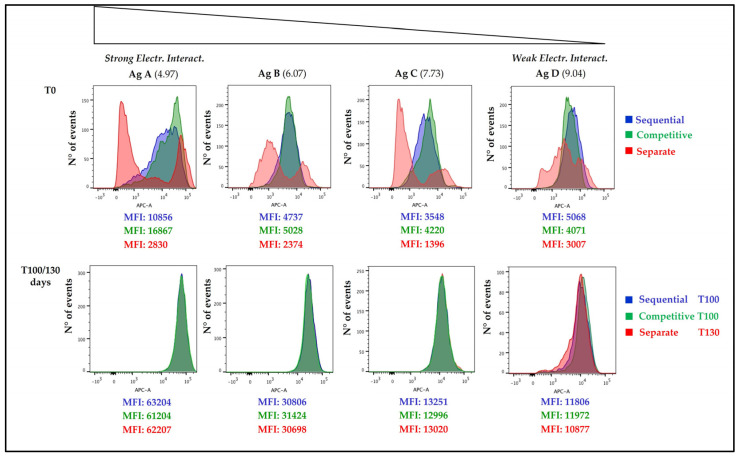
Fluorescence histogram at T0 and after 100/130 days of different formulation approach samples are reported. The IEP of each antigen is reported in brackets. On the X axis Fluorescence Intensity is reported, while on the Y axis number of events is reported. All numbers refer to Mean Fluorescence Intensity (MFI) values.

**Figure 7 vaccines-11-00155-f007:**
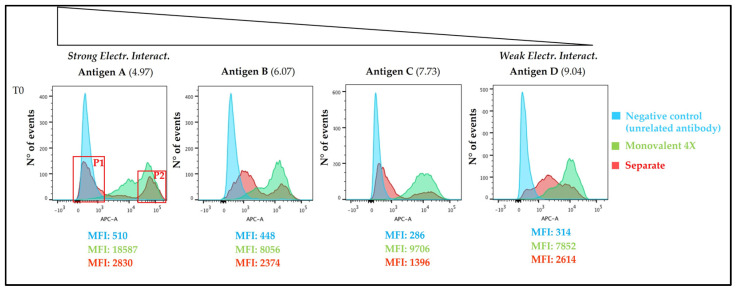
Fluorescence histogram at T0 of separate formulation approach sample (red) in comparison with negative control (light blue) and monovalent formulation (light green) are reported. On the X axis Fluorescence Intensity is reported, while on the Y axis number of events is reported. All numbers refer to Mean Fluorescence Intensity (MFI) values.

**Figure 8 vaccines-11-00155-f008:**
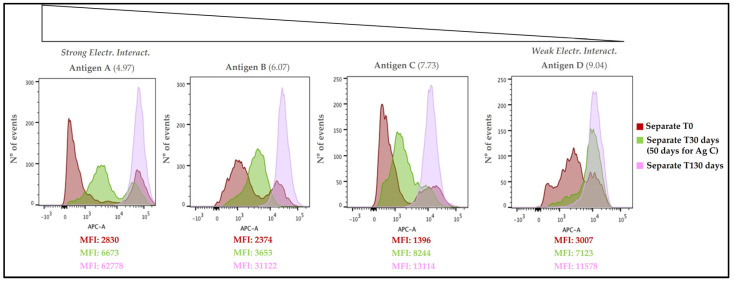
Separate adsorption sample, histogram overlay of time points 0-30-130 days for antigen A, B and D, while of 0-50-130 days for antigen C. On the X axis Fluorescence Intensity is reported, while on the Y axis number of events is reported. All numbers refer to Mean Fluorescence Intensity (MFI) values.

**Figure 9 vaccines-11-00155-f009:**
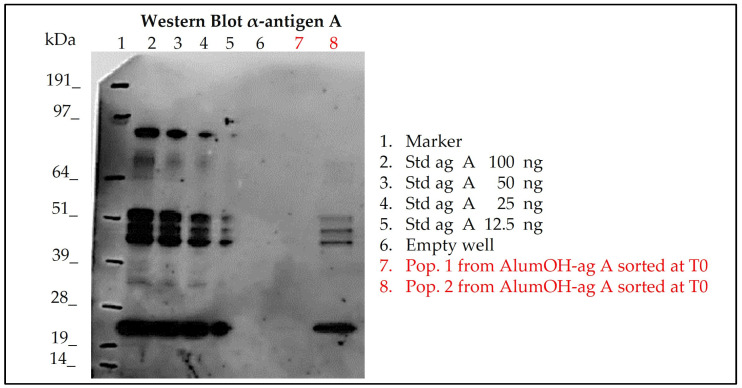
Western blot results of sorted AlumOH populations P1 and P2 after Flow Cytometry staining for antigen A in separate adsorption sample at T0, for orthogonal evaluation of delivered antigen content. Std ag: Standard antigen A solution of known amount used as references; P1 and P2: AlumOH populations sorted from separate adsorption sample at T0 after antigen A staining (see Figure 7).

**Table 1 vaccines-11-00155-t001:** Molecular weight and IEP of the antigens used in the study. The “supposed adsorption strength” is here classified based on the difference between AlumOH PZC (11.4) and antigen IEP, with the assumption of the greater the difference, the stronger the interaction. However, other mechanisms including for instance net charge at different pH, hydrogen, and hydrophobic bonding, van der Waals’ forces can play a role in adsorption.

Antigen Id.	Molecular Weight (Da)	Isoelectric Point (IEP)	Supposed Adsorption Strength
A	22,813	4.97	Strong
B	27,301	6.07	Intermediate
C	33,525	7.73	Intermediate
D	32,425	9.04	Weak

**Table 2 vaccines-11-00155-t002:** Overwiew of attributes tested, methods applied and expected data ranges.

Attribute	Method	Expected Result
pH	pH-metry	6.5 ± 0.5 (Histidine buffer)
AlumOH aggregation profile	Static Light Scattering	1–20 μm
Antigen integrity and adsorption	SDS-PAGE	No degradation and almost complete adsorption
AlumOH surface charge	Zeta Potential	≤0 (mV)
Antigen distribution	Flow Cytometry	Variation of Mean Fluorescence Intensity (MFI)

## Data Availability

The data presented in this study are available on request from the corresponding author.

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
