# Peer review of "Maturation of Aluminium Adsorbed Antigens Contributes to the Creation of Homogeneous Vaccine Formulations"

_vaccines, 2023, doi:10.3390/vaccines11010155_

Round 1
Reviewer 1 Report
1. This manuscript describes studies about antigens adsorption onto alum hydroxide used as a vaccine adjuvant. The study of antigen adsorption is an essential part of the development of formulations and must necessarily be done when developing any vaccine that uses adjuvants based on aluminum salts. the knowledge of the physical-chemical characteristics is of particular importance since it can represent the success or failure of an antigen with many potentials. For that, this manuscript brings some contributions.
2. As a strength, it can be said that the article contributes to the field of vaccine formulation, by exploring the interaction between antigens and adjuvant. The weakness lies in being a study with a certain limitation from the point of view of novelty and it reaffirms what people who work with vaccine formulations already know: the need to characterize the protein/aluminum hydroxide gel interaction.
3 I only have minor recommendations:
- Please check units and symbols all over the text. For instance, line 156, kg instead of Kg;
- Item 2.5 - SDS Page - please, indicate the type of gel staining;
- Lines 265 and 266 - please add one sentence explaining the reasons why samples after 50 and 130 days were not assayed for PSD
Author Response
Dear Reviewer,
Thanks for your comments. In the text we have modified the units and symbol Kg-mL-µL, as for your recommendation, respectively in kg-ml-µl.
At the end of section 2.5 we have added the following sentence: "The gel was stained using SimplyBlue™ SafeStain from Thermo Fisher Scientific, P/N LC6065, according to provider’s instructions".
Besides, in the section 3.1 after the statement “Formulation samples after 50 and 130 days were not assayed for PSD”, the following sentence has been added : “in order to limit sample consumption since the assay required the largest volume (about 500 µl)”. Please, consider that this decision was taken during stability/kinetic study since we did not expected to need a so long time point (130 days) to reach a complete uniform antigen distribution over time and also because there were not major changes in PSD values.
Warm regards
Donatello Laera
Reviewer 2 Report
The authors examined and compared the effects of three formulation strategies (sequential, competitive and separate antigen addition) on the formulation changes of four antigens loaded by aluminium oxyhydroxide over time. The results showed that antigen distribution across aluminium particles is a dynamic process that evolves over time. It is initially affected by the formulation approach and adsorption strength between antigen and the carrier, and eventually leads to homogeneous formulations. This research is of great significance for the aluminium adsorbed antigen formulation.
The authors characterized particle size distribution, surface charge, antigen integrity and adsorption on AlumOH, as well as antigen distribution by using the approaches including SDS-PAGE, flow cytometry and Western blot etc, and the above parameters were examined over a period of 130 days. The investigation was performed logically and the obtained data supported the conclusion. The discussion was appropriate. The manuscript was also well-written.
In my opinion, the manuscript can be published at its current form just from the academic point of view.
Author Response
The authors thanks the reviewer for the positive comment.
Warm regards
Donatello Laera